# Frequency of the Main Human Leukocyte Antigen A, B, DR, and DQ Loci Known to Be Associated with the Clearance or Persistence of Hepatitis C Virus Infection in a Healthy Population from the Southern Region of Morocco: A Preliminary Study

**DOI:** 10.3390/diseases12050106

**Published:** 2024-05-16

**Authors:** Safa Machraoui, Khaoula Errafii, Ider Oujamaa, Moulay Yassine Belghali, Abdelmalek Hakmaoui, Saad Lamjadli, Fatima Ezzohra Eddehbi, Ikram Brahim, Yasmine Haida, Brahim Admou

**Affiliations:** 1Laboratory of Immunology and HLA, Center of Clinical Research, Mohammed VI University Hospital, Marrakech 40080, Morocco; ideroujamaa@edu.uca.ma (I.O.); ab.hakmaoui@chumarrakech.ma (A.H.); saad.lamjadli@edu.uca.ma (S.L.); f.eddehbi.res@uca.ac.ma (F.E.E.); ikrambrahim@gmail.com (I.B.); yasminehaida188@gmail.com (Y.H.); br.admou@uca.ac.ma (B.A.); 2Biosciences Research Laboratory, Faculty of Medicine and Pharmacy, Cadi Ayyad University, Marrakech 40080, Morocco; 3African Genome Center, Mohammed VI Polytechnic University (UM6P), Ben Guerir 43151, Morocco; khaoula.errafii@um6p.ma; 4Department of Biology, Faculty of Sciences Dhar El Mahraz, Sidi Mohammed Ben Abdellah University, Fez 30003, Morocco; yassine.belghali@gmail.com

**Keywords:** HLA, frequency, HCV, healthy population, Moroccan population

## Abstract

Hepatitis C Virus (HCV) infection represents a significant global health challenge, with its natural course largely influenced by the host’s immune response. Human Leukocyte Antigen (HLA) molecules, particularly HLA class I and II, play a crucial role in the adaptive immune response against HCV. The polymorphism of HLA molecules contributes to the variability in immune response, affecting the outcomes of HCV infection. This study aims to investigate the frequency of HLA A, B, DR, and DQ alleles known to be associated with HCV clearance or persistence in a healthy Moroccan population. Conducted at the University Hospital Center Mohammed VI, Marrakech, this study spanned from 2015 to 2022 and included 703 healthy Moroccan individuals. HLA class I and II typing was performed using complement-dependent cytotoxicity and polymerase chain reaction-based methodologies. The results revealed the distinct patterns of HLA-A, B, DRB1, and DQB1 alleles in the Moroccan population. Notably, alleles linked to favorable HCV outcomes, such as HLA-DQB1*0301, DQB1*0501, and DRB1*1101, were more prevalent. Conversely, alleles associated with increased HCV susceptibility and persistence, such as HLA-DQB1*02 and DRB1*03, were also prominent. Gender-specific variations in allele frequencies were observed, providing insights into genetic influences on HCV infection outcomes. The findings align with global trends in HLA allele associations with HCV infection outcomes. The study emphasizes the role of host genetics in HCV infection, highlighting the need for further research in the Moroccan community, including HCV-infected individuals. The prevalence of certain HLA alleles, both protective and susceptibility-linked, underscores the potential for a national HLA data bank in Morocco.

## 1. Introduction

Hepatitis C Virus (HCV) infection is characterized by inflammation that can potentially result in severe liver damage [1]. Identified as a significant global health threat, the World Health Organization (WHO) estimates that 58 million people are living with chronic HCV, making it the primary cause of liver cancer, with approximately 290,000 deaths annually [2].

The immune response against HCV plays a pivotal role in shaping the natural course of the infection [3,4]. As the key components of the immune system, Human Leukocyte Antigen (HLA) or Major Histocompatibility Complex (MHC) molecules play a crucial role in the adaptive immune response against HCV infection [3]. HLA molecule polymorphism enables the immune system to identify a diverse range of antigens, including viral proteins derived from HCV [5]. HLA class I and II molecules are responsible for presenting HCV antigens to CD8+ and CD4+ T cells [6], initiating specific immune responses to eliminate infected cells [3]. However, the effectiveness of HLA molecules in recognizing HCV antigens and initiating an immune response varies among individuals, thereby influencing the natural history of the infection [5].

Numerous studies have explored the role of HLA molecules in the clearance or persistence of HCV infection [7]. Indeed, strong and effective CD4+ helper and CD8+ effector T cell responses are associated with HCV clearance [8], while the functional exhaustion of HCV-specific CD4+ and CD8+ T cells has been linked to the progression to chronicity [9,10]. Certain HLA alleles are linked to more effective immune responses against HCV, resulting in viral clearance [11,12]. Individuals carrying HLA-DQB1*0301, DQB1*0501, DRB1*1101, A*03, and B*27 are shown to have a higher likelihood of developing a strong CD8+ T cell response against HCV, leading to viral control and spontaneous clearance [13,14,15]. Conversely, other alleles are associated with a higher risk of chronic HCV infection, such as HLA-DQB1*0201, DRB1*0301, DRB1*0701, and DQB1*0401, indicating increased susceptibility to viral persistence [16,17,18]. Furthermore, some HCV variants can escape recognition by CD8+ T cells when viral particles are presented through specific HLA molecules, promoting chronic infection [19].

According to the literature, the frequency and distribution of the different HLA specificities vary depending on the ethnic origin of the populations [20,21,22,23,24]. Such variation can impact the global epidemiology and clinical expression of the HCV infection in terms of severity and complications [25]. To the best of our knowledge, studies focusing on the link between the distribution of HLA molecules in the general population and their impact on the HCV natural history in the Moroccan population are scarce. The objective of this study was to determine the frequency of the main HLA A, B, DR, and DQ loci known to be associated with the clearance or persistence of HCV infection in a healthy population from the southern region of Morocco.

## 2. Material and Methods

### 2.1. Study Design

This investigation was conducted in the HLA Laboratory of the Clinical Research Center (CRC) at the University Hospital Center Mohammed VI, Marrakech. The retrospective study focused on a cohort of Moroccan participants, who were healthy individuals predominantly residing in the southern regions of Morocco, encompassing Marrakech-Safi, Souss-Massa, Beni-Mellal Khenifra, Guelmim Oued Noun, and Draa-Tafilalt governorates. The recruitment spanned from 2015 to 2022, culminating in the inclusion of 703 healthy Moroccan individuals. Among these, 684 underwent HLA class I typing, while 323 were subjected to HLA class II typing.

The inclusion criteria were meticulously established through a thorough review of the participants’ medical histories, ensuring their overall health and the absence of positive viral serologies, specifically HIV, HBV, and HCV. Exclusion criteria were stringently applied to omit recipients awaiting organ, hematopoietic stem cell (HSC), or bone marrow transplants, as well as potential donors who exhibited positive HCV serology.

### 2.2. Ethical Considerations

The samples analyzed in this study were obtained as a part of the routine activities of the HLA laboratory of Mohammed VI University Hospital in Marrakech. Sociodemographic data were extracted from the computer database anonymously under the supervision of the laboratory manager. In this case, approval of ethics and informed consent were not necessary.

### 2.3. Sample Collection and Processing

Peripheral venous blood was collected adhering to a rigorously defined protocol designed for Human Leukocyte Antigen (HLA) typing. For HLA class I allele typing, we utilized two 9 mL Lithium Heparin tubes. In parallel, HLA class II allele typing necessitated the use of two 5 mL tubes containing Ethylenediaminetetraacetic Acid (EDTA) as an anticoagulant. This bifurcated approach ensures the stabilization and preservation of different cell types pertinent for comprehensive HLA typing. Upon collection, the samples were immediately and carefully transported to the processing laboratory. Transportation was conducted under controlled conditions using specialized sample transport bags, designed to maintain sample integrity during transit. This immediate transfer is critical to prevent degradation and to ensure the preservation of the cellular components necessary for accurate HLA typing.

DNA extraction: Genomic DNA extraction from peripheral blood mononuclear cells (PBMCs) was executed using the QIAmp DNA Mini kit (Qiagen, Hilden, Germany), following a multi-step protocol. Briefly, this process began with cell lysis, where PBMCs were treated with a lysis buffer containing chaotropic salts to disrupt cell membranes and release genomic DNA. The lysate was then applied to a silica-based column, where the genomic DNA was selectively bound to the silica membrane, allowing impurities to be washed away. A series of ethanol-based wash buffers were used to cleanse the DNA, ensuring the removal of salts, metabolites, and other contaminants. Finally, the purified DNA was eluted from the membrane using an elution buffer, carefully releasing the DNA without causing damage. The resulting DNA was of high quality, as confirmed by quantitative and qualitative assessments using a NanoDrop^TM^ 2000/2000c Spectrophotometer (Thermo Scientific™, Waltham, MA, USA), which measured DNA concentration and purity, crucial for downstream molecular analyses.

### 2.4. HLA Typing Methodology

Human Leukocyte Antigen (HLA) typing for class I (HLA-A and HLA-B) alleles was performed employing the complement-dependent cytotoxicity (CDC) method, utilizing reagents and protocols provided by OneLambda™, Los Angeles, CA, USA. This traditional and highly specific method involves the incubation of patient lymphocytes with specific anti-HLA sera, followed by the addition of complement. Cell death, indicative of a match between the serum antibody and the HLA antigen on the lymphocyte, is then assessed microscopically, allowing for the precise HLA allele identification.

In parallel, HLA class II allele typing (focusing on DRB1 and DQB1 loci) was executed using two distinct polymerase chain reaction (PCR)-based methodologies, catering to the need for high specificity and sensitivity. The first approach utilized was the Sequence-specific Primer (SSP) technique, again facilitated by OneLambda™ CA, USA. This method involves PCR amplification using primers specific to particular HLA sequences, with the resulting amplicons being subjected to electrophoretic analysis for allele determination (Figure 1). The second approach employed was the Sequence-specific Oligonucleotide (SSO) typing, provided by Immucor™, Peachtree Corners, GA, USA. This technique incorporates PCR amplification followed by the hybridization of the amplified products to oligonucleotide probes that are specific to known HLA sequences. The hybridization patterns are then analyzed, providing a comprehensive profiling of the HLA class II alleles.

Both methods for HLA class I and II typing were rigorously performed under controlled conditions, ensuring accurate and reliable typing, which is pivotal for various clinical applications such as transplantation compatibility assessment and disease association studies.

### 2.5. Statistical Analysis

The primary focus of the analysis was on determining the allelic frequencies within the study population, a key metric in genetic epidemiology and population genetics studies. Allelic frequency, in this context, refers to the relative frequency of a specific allele (variant of a gene) within the study group. To calculate this, the frequency was determined as the proportion of each individual allele in comparison to the total number of alleles present in the sample population. Specifically, this was achieved by dividing the number of occurrences of the allele of interest (the test allele) by the total number of alleles observed across all individuals in the group. This method of calculating allelic frequency is critical for understanding the distribution of genetic traits in the population and can provide insights into various genetic predispositions and potential associations with diseases or other phenotypic characteristics. The use of IBM SPSS Statistics 20 software in this process ensures precise calculations and reliable statistical analysis, enabling a robust interpretation of the genetic data collected in the study.

## 3. Results

The laboratory characteristics of the studied groups are shown in Table 1.

### 3.1. Frequencies of HLA Class I (A and B) in the Study Population

In this study, we delve into the analysis of HLA class I loci A and B in a healthy Moroccan population. The primary objective is to present the results of allele frequency, offering a comprehensive perspective on genetic diversity within this specific demographic. This analysis is a part of a broader investigation aimed at understanding the genetic underpinnings of clearance or persistence of HCV infection along with immune response in the Moroccan population.

In the detailed analysis of HLA-A typing among a sample of 684 Moroccan individuals, the distribution of specific HLA-A alleles revealed a distinct pattern. The HLA-A2 allele emerged as the most prevalent, observed in 23.3% of the individuals. This was followed by the HLA-A1 allele, present in 11.6% of the cohort. The third most common allele was HLA-A3, found in 8.9% of the participants. Notably, two alleles, HLA-A23 and HLA-A24, shared equal prevalence, each accounting for 8% of the population. Coming close were the HLA-A68 and HLA-A30 alleles, found in 7.8% and 6.5% of the individuals, respectively (Table 2). This distribution underscores a diverse genetic composition within the studied group, reflecting the unique HLA allele frequencies in the Moroccan population. Such insights are crucial for understanding genetic predispositions and immune response variations in this demographic. Building upon the insights from the HLA-A allele analysis in the Moroccan cohort, we extend our focus to the HLA-B locus to further comprehend the genetic diversity within this population. In the same subset of 684 individuals, we conducted HLA-B typing and identified a total of 1368 alleles, highlighting the extensive polymorphism characteristic of the HLA-B locus. The HLA-B51 allele emerged as the most predominant in this group, representing 9.9% of the alleles. This was closely followed by the HLA-B44 allele, accounting for 9.1% of the population. Other notable alleles included HLA-B8, HLA-B49, and HLA-B45, with frequencies of 6.7%, 6.4%, and 6%, respectively (Table 2). These findings not only reinforce the genetic diversity observed in the HLA-A locus analysis but also emphasize the unique allele distribution patterns within the Moroccan population. Such detailed understanding of HLA-B allele frequencies is vital for a range of applications, from disease association studies to transplant compatibility assessments [26], enhancing our capability to tailor medical interventions to the genetic profile of this specific population.

### 3.2. Frequencies of HLA Class II (DRB1 and DQB1) Loci in the Study Population

Our study further delves into the genetic landscape of the Moroccan population by examining the HLA class II loci, specifically DRB1 and DQB1 (Table 3). This analysis is crucial, as class II HLA molecules play a pivotal role in the immune system’s regulation, particularly in orchestrating the responses of CD4+ T helper cells [27,28]. The DRB1 and DQB1 loci are known for their genetic variability and are often implicated in various autoimmune and infectious diseases, making them highly relevant for understanding disease susceptibility and immune response mechanisms [5]. In our cohort of 323 individuals, we undertook comprehensive HLA-DRB1 and HLA-DQB1 typing. This subset of the study provided 646 alleles, offering a detailed view of the allelic distribution within these critical loci. In the HLA-DRB1 locus analysis, we identified the HLA-DRB1*03 allele as the most frequent, present in 19.2% of the alleles. This was closely followed by the HLA-DRB1*13 and DRB1*07 alleles, with two distinct frequencies: 15.8% and 14.9%. Another significant finding was the HLA-DRB1*15 allele, constituting 13.3% of the alleles. Turning to the HLA-DQB1 locus, the study revealed that the HLA-DQB1*02 allele was the most prevalent, substantially observed in 33.1% of the alleles. Other prominent alleles in this group included HLA-DQB1*06 at 24.3%, HLA-DQB1*03 at 24%, and HLA-DQB1*05 at 13%. The focus on these specific loci is driven by their critical role in immune response regulation and their association with various health conditions [22]. The findings from this analysis not only contribute to the global understanding of HLA diversity but also provide a valuable genetic framework specific to the Moroccan population, which can be instrumental in guiding healthcare strategies, including HCV infection prevention, diagnosis, and treatment.

### 3.3. Frequencies of HLA Class I and II Allele Groups Associated with Hepatitis C Infection Outcomes

In this study, we have identified various (HLA) alleles and their frequencies, which are potentially associated with differing outcomes for the HCV infection. This identification is grounded in the existing literature, which suggests certain HLA alleles might influence the course and outcome of the HCV infection [13,14,15,16,18,29]. Notably, the alleles HLA-DQB1*03, DRB1*07, DQB1*05, DRB1*11, A03, and B27 were observed at frequencies of 24%, 13%, 10.1%, 8.9%, and 3.7%, respectively. These alleles are of interest because they have been linked in previous research to favorable outcomes in the HCV infection, such as spontaneous viral clearance or a more effective immune response [13,14,15].

Conversely, we also identified alleles that are reportedly associated with increased susceptibility to HCV infection and the likelihood of viral persistence. These include HLA-DQB1*02, DRB1*03, DRB1*07, and DRB1*04, which were found at frequencies of 33.1%, 19.2%, 14.9%, and 12.7%, respectively. The presence of these alleles in the population could imply a higher risk for chronic HCV infection, as suggested by the literature [16,18,29].

### 3.4. Distribution of HLA-A and B Loci Allele Groups According to Gender

For the female population, the predominance of the HLA-A2 allele group at 23.6% indicates a significant representation of this allele. Following HLA-A2, the HLA-A1 allele was present in 12.31% of females, suggesting its considerable occurrence in the population. Other alleles, including HLA-A3 (9.42%), HLA-A68 (8.05%), HLA-A24 (7.6%), and HLA-A23 (7%), also showed noteworthy frequencies. These patterns highlight a diverse HLA-A allele profile among females, which could be influential in understanding gender-specific immune responses and disease susceptibilities. Conversely, the male population demonstrated a slightly different HLA-A allele distribution. The HLA-A2 allele, similar to the female cohort, was the most common but at a slightly lower frequency of 23.1%. This was followed by HLA-A1 at 11%, indicating its notable presence but less than that in females. The HLA-A23 allele was more prevalent in males (8.87%) compared to females, which might suggest a gender-specific genetic predisposition. Other alleles such as HLA-A3 (8.45%), HLA-A24 (8.31%), and HLA-A30 (7.61%) also showed significant frequencies in the male population (Appendix A).

For the female population, the HLA-B51 allele group emerged as the most prevalent, found in 10.94% of the cohort. This was closely followed by the HLA-B49 allele at 8.21%, demonstrating its significant presence among females. The HLA-B44 allele was also notable, comprising 8% of the female group. Additionally, HLA-B8 and HLA-B35 were present at 7.29% and 6.38%, respectively. This diverse range of HLA-B alleles in females may have important implications for understanding gender-specific immune mechanisms and HCV susceptibilities.

Conversely, in the male population, the HLA-B44 allele group was more dominant, observed in 10.28% of males. This was closely followed by HLA-B51 at 9%, indicating a slightly different distribution pattern compared to females. Other significant alleles in males included HLA-B7 at 6.76% and HLA-B45 at 6.2% (Appendix A). The variation in the prevalence of these HLA-B alleles between genders underscores the importance of considering gender differences in genetic research, particularly for HCV where the immune response plays a crucial role.

### 3.5. Distribution of HLA-DRB1 and DQB1 Allele Groups According to Gender

In the Moroccan population, our study revealed distinct gender-specific distributions of HLA-DRB1 and HLA-DQB1 alleles. Among females, HLA-DRB1*03 was the most common allele at 20.18%, followed by HLA-DRB1*07, HLA-DRB1*13, and HLA-DRB1*04. In contrast, males showed a slightly different pattern with HLA-DRB1*03, leading at 18.15%, and closely followed by HLA-DRB1*13 and HLA-DRB1*15. Regarding the HLA-DQB1 locus, females predominantly had HLA-DQB1*02 at 36.14%, whereas in males, this allele was also prevalent but at a lower frequency of 29.94% (Appendix A).

### 3.6. Frequencies of HLA Class I and II Alleles Associated with Outcomes of Hepatitis C Infection According to Gender

In our study, we explored the prevalence of specific Human Leukocyte Antigen (HLA) alleles associated with Hepatitis C Virus (HCV) outcomes in both female and male participants. Notably, among the alleles linked to viral control and spontaneous clearance, we observed gender-specific variations. HLA-DQB1*03, HLA-DQB1*05, HLA-DRB1*11, HLA-A03, and HLA-B27 exhibited distinct frequencies. In females, these alleles were found at frequencies of 24.1%, 13.25%, 10.84%, 16.87%, 8.9%, and 2.74%, respectively, while in males, the frequencies were 23.89%, 12.74%, 9.24%, 12.74%, 8.45%, and 4.51%. Additionally, alleles associated with susceptibility and viral persistence, including HLA*-*DQB1***02, HLA-DRB1*03, HLA-DRB1*07 and HLA-DRB1*04, demonstrated varying frequencies in both genders. Females exhibited frequencies of 36.14%, 20.18%, and 12.35%, while males showed frequencies of 29.94%, 18.15%, and 13.06%, respectively. These gender-specific allele distributions offer valuable insights into the genetic factors influencing HCV infection outcomes (Appendix A).

## 4. Discussion

HLA class I and II molecules play a significant role in shaping the host’s immune response to viral infections by serving as crucial proteins for presenting antigens to CD8+ and CD4+ T helper cells [30,31]. This study, conducted in a healthy population from the southern region of Morocco, aimed to investigate the frequency of HLA class I and II alleles associated with the outcomes of HCV infection. The DQB1*0301 allele consistently plays a role in the natural elimination of HCV infection across various populations, as evidenced by multiple studies [21,22,23,32]. Investigations carried out in individuals from France [33], Egypt [21], and Turkey [34], each with diverse genetic backgrounds, have revealed a notable high frequency of HLA-DQB1*03 and -DRB1*11 among healthy controls. The link between the DRB1*0101 and DQB1*0501 alleles and the clearance of the virus has been observed among Caucasian Americans. However, among African American patients, viral clearance is primarily linked with the DQB1*0301 allele [16]. Our series comprised Moroccan individuals selected from the broader population of the same geographical region. Some globally recognized alleles associated with viral control and spontaneous clearance of HCV infection, such as HLA-DQB1*03, DQB1*05, DRB1*07, and DRB1*11, appeared more frequently in our population compared to other specificities, particularly HLA-DQB1*03. These results align with previous findings, and among Egyptian healthy control group, DRB1*11:01:01 was the most frequent allele (14.88%) followed by DRB1*3:01:01 (13.1%) and DRB1*07:01:01 (10.88%) [21]. These alleles were associated with protection against the HCV infection [15,21,22,23,32]. On the contrary, our population demonstrated the dominance of HLA-DQB1*02 and DRB1*03, which are highly linked to HCV susceptibility and persistence of the infection [11,15,20,21]. The high prevalence of these specificities among the Moroccan population [26,27] may account for their high frequency in our healthy cohort. These genetic associations offer valuable insights into the role of host genetics in determining HCV infection outcomes. Further research, specifically including individuals infected with HCV in the Moroccan community, is needed.

HLA class I polymorphism appears to be less studied in HCV infection outcomes according to the literature [7]. Protection from HCV infection has been associated with B*35 in the Tunisian population [23], while in Egypt, researchers found that B*50 was strongly linked to spontaneous clearance of HCV infection [21]. Common HLA class I alleles associated with either the protection or susceptibility to HCV infection across different populations are rarely observed [7]. However, in Irish and German populations, HLA-A*03 and B*27 alleles were most strongly associated with HCV clearance [4,28].

HLA-A2, -A1, -B51, and –B44 are the most frequent HLA class I genes among our healthy population. These findings align with other studies conducted in healthy Moroccan population by Brick et al. and Kabbaj et al., who reported that HLA-A1, -A2, and –B44 are the most frequent loci [35,36,37]. HLA genes are linked to persistence of the HCV infection, particularly HLA-A28 and -B14 in the Egyptian population [21], -A19 in the Saudi Arabian population [38], and -B38 in the Turkish population [39]. The presence of a link between HLA-A3 or other genes and protection against HCV infection or susceptibility in our healthy population needs further confirmation by investigating the frequency of these genes in an HCV-infected population.

Considering the gender of our population, either for HLA class I or class II, there was no significant difference in terms of alleles frequency regardless of whether the alleles are linked to HCV infection or not.

## 5. Conclusions

This preliminary study provides insight into the distribution of the main HLA A, B, DR and DQ loci associated with HCV infection in the healthy population of southern Morocco, notably, the high frequency of HLA-DQB1*03, DQB1*05, and DRB1*11 alleles, considered protective against chronic infection, contrasting with the low frequency of HLA-DQB1*04 as a predisposing allele. These data could contribute to understanding the influence of immunogenetic susceptibility to HCV infection in our context. These alleles can be considered as the genetic determinants of HCV infection outcomes and may provide novel insights into HCV pathophysiology, hence, offering promising avenues for HCV management.

These findings also underline, the importance of establishing a national HLA data bank for the Moroccan population, which could facilitate HLA–disease association studies, as well as anthropological research in our context.

Further research encompassing larger cohorts with HCV infected patients and comprehensive genetic analyses is warranted to validate and extend these initial findings, potentially informing strategies for personalized healthcare and therapeutic interventions tailored to the diverse genetic backgrounds of populations at risk of HCV.

## Figures and Tables

**Figure 1 diseases-12-00106-f001:**
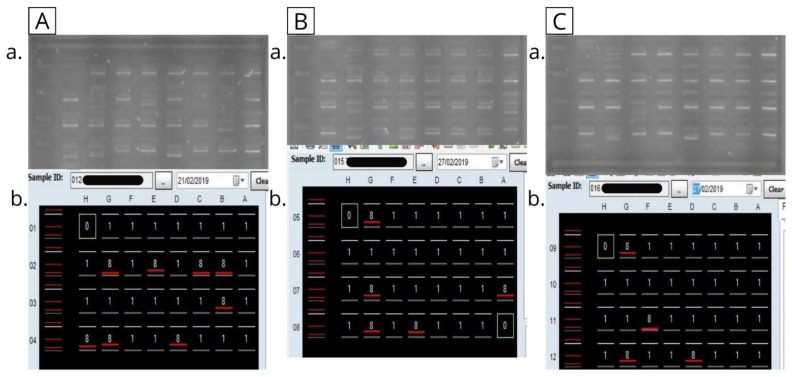
Results of PCR-SSP genotyping (intermediate-resolution) for HLA-DRB and -DQB loci of 3 cases. The PCR products are loaded into each well of 2% agarose gel stained with ethidium bromide, electrophoresed, and visualized under UV light. 0 = negative control; 8 = positive reaction; and 1 = negative reaction. The band present in all lanes except the negative control is the positive internal control amplicon, and the positive band is located beyond the internal control band which is indicated by the red line (score 8) in the correspondent image below, while negative band is indicated by score 1. The interpretation and assignment of HLA loci was carried out using HLA Fusion software (One Lambda). Examples of PCR-SSP typing results are shown in images (**A**–**C**), coresponding to HLA-DQB1*03, DQB1*05; DRB1*11, *DRB1*14, and DRB1*13 loci (**case 012, A.a and A.b**); to HLA-DQB1*02, DQB1*05; DRB1*01, and *DRB1*07 loci (**case 015, images B.a and B.b**) and to HLA-DQB1*03, DQB1*05; DRB1*01, and *DRB1*08 loci (**case 016, images C.a and C.b**).

**Table 1 diseases-12-00106-t001:** Demographic characteristics of healthy groups.

Locus	Sex Ratio	Age Average
locus A		
684 (353 male, 331 females)	1.066	28.76
locus B		
684 (353 male, 331 females)	1.066	28.76
Locus DRB1		
323 (157 males, 166 females)	0.94	29.39
Locus DQB1		
323 (157 males, 166 females)	0.94	29.39

**Table 2 diseases-12-00106-t002:** Frequency of HLA class I (A and B) in the study population.

HLA Class I			
HLA-A Locus	Frequency (%)	HLA-B Locus	Frequency (%)
A2	23,3	B51	9.9
A1	11.6	B44	9.1
A3	8.9	B8	6.7
A23	8	B49	6.4
A24	8	B45	6
A68	7.8	B35	5.7
A30	6.5	B7	5.7
A33	4	B50	5.2
A11	3.7	B14	4.4
A32	3.7	B18	4.2
A29	3.1	B58	4.2
A26	2.6	B27	3.7
A34	2.3	B38	3.1
A80	1.5	B53	2.6
A31	1.2	B40	2.5
A66	1	B41	2.4
A28	0.8	B57	2.3
A74	0.8	B42	2.1
A10	0.4	B63	2.1
A36	0.2	B39	2
A69	0.2	B15	1.7
A9	0.2	B72	1.7
A19	0.1	B52	1.2
A25	0.1	B13	1
		B17	0.8
		B37	0.7
		B78	0.7
		B47	0.4
		B21	0.3
		B55	0.2
		B56	0.2
		B62	0.2
		B12	0.1
		B2	0.1
		B64	0.1
		B65	0.1
		B70	0.1
		B71	0.1
		B73	0.1

Note: This series may contain broads and splits due to the limits of the methods used for HLA class I typing, notably the CDC method.

**Table 3 diseases-12-00106-t003:** Frequency of HLA DRB1 and DQB1 loci in the study population.

HLA Class II			
HLA-DRB1 Locus	Frequency (%)	HLA-DQB1 Locus	Frequency (%)
DRB1*03	19.2	DQB1*02	33.1
DRB1*13	15.8	DQB1*06	24.3
DRB1*07	14.9	DQB1*03	24
DRB1*15	13.3	DQB1*05	13
DRB1*04	12.7	DQB1*04	5.4
DRB1*11	10.1	DQB1*07	0,2
DRB1*01	7.4		
DRB1*08	2.5		
DRB1*09	1.5		
DRB1*10	0.9		
DRB1*14	0.9		
DRB1*12	0.6		
DRB1*16	0.2		

## Data Availability

The data used in this study are not publicly available due to privacy restrictions. Access to the data is restricted in accordance with the ethical guidelines and regulations governing the protection of participant confidentiality and privacy. However, researchers interested in replicating or verifying the findings presented in this study may request access to the data through the appropriate institutional review board. Requests for data access will be considered on a case-by-case basis, subject to approval by the relevant authorities and compliance with applicable privacy regulations. For inquiries regarding data access, please contact Pr. Brahim ADMOU/Clinical research Center /Mohammed VI University Hospital Center at br.admou@uca.ac.ma.

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
