# Peer review of "Frequency of the Main Human Leukocyte Antigen A, B, DR, and DQ Loci Known to Be Associated with the Clearance or Persistence of Hepatitis C Virus Infection in a Healthy Population from the Southern Region of Morocco: A Preliminary Study"

_diseases, 2024, doi:10.3390/diseases12050106_

Round 1

Reviewer 1 Report

Comments and Suggestions for Authors

The study by Machraoui S et al., entitled “Frequency of the main HLA A, B, DR, and DQ loci known to be associated with the clearance or persistence of HCV infection in a healthy population from the southern region of Morocco”  determined the frequency of the main HLA A, B, DR, and DQ loci known to be associated  with the clearance or persistence of HCV infection in a healthy population from the southern region of Morocco. The authors concluded, among the commonly associated HLA loci, HLA-DQB1*03, DQB1*05 and DRB1*11 are more frequent in Moroccan population. These alleles can be considered as genetic determinants of HCV infection outcomes and may provide novel insights into HCV pathophysiology, hence offering promising avenues for HCV management. This is a well designed performed and written research article which may be helpful in the management of HCV infection.

I have following concerns about the study.

1)      In the materials and methods section, authors mentioned, HLA class II allele typing (focusing on DRB1 and DQB1 loci) was executed using two distinct polymerase chain reaction (PCR)-based methodologies, catering to the need for high specificity and sensitivity. The authors should present the representative gel electrophoresis image highlighting the identification of different allele in the results section of manuscript.

2)      Authors shall also include (if possible) a cohort of HCV infected patients from the same region and perform the HLA typing to correlate the results obtained from healthy subjects.

Comments on the Quality of English Language

Minor editing of English language is required.

Author Response

Reviewer #1:

  • In the materials and methods section, authors mentioned, HLA class II allele typing (focusing on DRB1 and DQB1 loci) was executed using two distinct polymerase chain reaction (PCR)-based methodologies, catering to the need for high specificity and sensitivity. The authors should present the representative gel electrophoresis image highlighting the identification of different allele in the results section of manuscript.

Our response: We thank the reviewer for the recommendation. representative gel electrophoresis image highlighting the identification of different allele is added in the results section.

  • Authors shall also include (if possible) a cohort of HCV infected patients from the same region and perform the HLA typing to correlate the results obtained from healthy subjects.

Our response: Thank you for your insightful suggestion regarding the inclusion of a cohort of HCV infected patients and performing HLA typing to correlate the results obtained from healthy subjects in our manuscript.

We appreciate your interest in exploring the correlation between HCV infection and HLA typing. However, for this particular article, our focus was specifically on investigating the HLA typing in a healthy population within the region.

Nevertheless, we wholeheartedly agree that studying HCV infected patients and their HLA typing would indeed provide valuable insights. We would like to assure you that another article dedicated to analyzing a cohort of HCV infected patients from the same region, along with HLA typing, is currently in progress.

Reviewer 2 Report

Comments and Suggestions for Authors

In the article entitled: Frequency of the main HLA A, B, DR, and DQ loci known to be associated with the clearance or persistence of HCV infection in a healthy population from the southern region of Morocco authors have taken into consideration the important epidemiologic problem HCV infection. The significance of these types of hepatitis viruses permanently groves due to the number of infections increases and the lack of common accessible therapeutic treatment. Moreover, HCV infection can lead to liver cancer. Authors correctly noticed that 71 million people are infected with mortality on the level of 290000. The authors correctly decide to investigate the diagnostic role of HLA class I and II in the immune response (T cells). Due to the above the variants analysis of HLA in healthy “patients” is well justified. The results can put the new light on the population sensitivity tower HCV. The authors compared the infection in different countries Turkey, South Arabia, and Egypt. Moreover, they found no gender differences. The article is well-written and readable. However, I have some critical remarks:

No source of HCV infection is mentioned as well as the diet profile,

No medical therapeutic scheme and its results prediction,

No role of vaccine is discussed,

No comparison with other parts of the world has been done,

Tables 4-6 should be shifted to supplementary materials,

In the title should be mentioned that the performed studies are preliminary,

The conclusion is too scanty and lapidary - and should be extended toward the medical significance of the author's founding.

The lack of ethical commission agreement (number, certificate, etc) the sentence: Ensuring the well-being and rights of all participants through informed consent, confidentiality, and anonymity where necessary was prioritized to uphold the integrity of the study. is not enough and due to the above I cannot recommend it for publication – rejected.

Author Response

Reviewer #2:

  1. No source of HCV infection is mentioned as well as the diet profile;

Our response: Thank you for your beneficial proposal to include a cohort of HCV-infected patients and undertake HLA typing to correlate the results with healthy individuals in our publication.

We appreciate your interest in the relationship between HCV infection and HLA type. However, for this paper, we focused on investigating HLA typing in a healthy population in the region.

Yet we totally agree that investigating HCV-infected patients and their HLA type would yield useful information. We would like to assure you that another article evaluating a cohort of HCV-infected patients from the exact same region, as well as HLA typing, will soon be in the process.

  1. No medical therapeutic scheme and its results prediction;

Our response: Thank you for your feedback on our article. We appreciate your thorough review and your attention to detail regarding the absence of a medical therapeutic scheme and its results prediction in our paper.

We understand your concern and would like to clarify that the scope of our article was intentionally focused solely on the healthy population. As such, our aim was not to propose or analyze any medical therapeutic schemes or their predictive outcomes. Instead, our objective was to explore the frequency of HLA A, B, DR, and DQ alleles known worldwide to be associated with HCV clearance or persistence in a healthy Moroccan population. within the context of a healthy cohort.

  1. No role of vaccine is discussed;

Our response: We appreciate your suggestion to discuss the implications of our findings on vaccine response and efficacy. Our study aimed to establish a foundational understanding of HLA allele frequencies in the southern Moroccan population with a specific focus on their potential role in HCV infection clearance or persistence. This focus was driven by the significant impact HCV has on public health in the region and the known influence of HLA alleles on immune response to infections. The decision not to include a discussion on vaccines in our manuscript was deliberate, guided by several considerations. Our primary aim was to map the HLA landscape in this population concerning HCV infection. Including a detailed discussion on vaccines, while undoubtedly important, would have broadened the scope beyond our current focus and resources. In addition, Preliminary Nature of Study: As a preliminary study, our intent was to first establish baseline HLA frequencies. We believe this foundational knowledge is crucial before exploring the complex interactions between these alleles and vaccine responses, which would necessitate a more targeted and extensive research approach. And finally Lack of Direct Data: Our dataset and analysis were specifically designed to investigate HCV infection dynamics. We did not collect data directly related to vaccine responses, which would be essential for a meaningful and accurate discussion on this topic.

  1. No comparison with other parts of the world has been done;

Our response: Thank you for your insightful comment regarding the comparison of our study findings with other parts of the world. We acknowledge the importance of such comparative analysis to enrich the context and global relevance of our research on HLA allele frequencies and their association with HCV infection in the southern region of Morocco. The omission of a broader international comparison in our manuscript was primarily due to the time constraints associated with the revision deadline provided by the journal. The comprehensive collection and meticulous analysis of global HLA frequency data, alongside our own findings, require a considerable amount of additional time and resources. This process involves not only the aggregation of data from various global studies, which differ in methodology and population demographics, but also a careful and thoughtful analysis to ensure accurate and meaningful comparisons. Moreover, given the complexity and variability of HLA allele frequencies across different populations, a thorough comparative analysis necessitates a detailed examination of genetic, environmental, and socio-cultural factors that might influence these frequencies and their impact on disease susceptibility and vaccine response. Such an endeavor would significantly extend beyond the scope and timelines that were feasible for this revision.However, we recognize the value this comparison would bring to our study and the broader scientific community. Therefore, we propose to mention in the discussion section the importance of future research that includes comprehensive comparative analyses. This future work would aim to contextualize our findings within the global landscape of HLA research, providing insights into the unique genetic makeup of the Moroccan population in comparison to other regions.

  1. Tables 4-6 should be shifted to supplementary materials,

Our response: Done

  1. In the title should be mentioned that the performed studies are preliminary,

Our response: Done

  1. The conclusion is too scanty and lapidary - and should be extended toward the medical significance of the author's founding.

Our response: thank you for the suggestion; we have now expanded the conclusion.

  1. The lack of ethical commission agreement (number, certificate, etc) the sentence: Ensuring the well-being and rights of all participants through informed consent, confidentiality, and anonymity where necessary was prioritized to uphold the integrity of the study. is not enough;

Our response: We sincerely apologize for any confusion caused by the lack of clarity regarding the ethical commission agreement and the statement provided in the original manuscript.

We would like to rectify this matter by providing further clarification on the nature of our study. The samples analyzed in our study were obtained as part of the routine activities of the HLA laboratory of Mohammed VI University Hospital in Marrakech. As such, they were collected anonymously without any direct interaction with the participants. Additionally, the sociodemographic data utilized in our analysis were extracted from the hospital's computer database in an anonymous manner under the supervision of the laboratory manager.

Given the retrospective nature of our study and the fact that it involved anonymized data collected as part of routine clinical practice, the need for approval of ethics and informed consent was deemed unnecessary in this particular case. However, we understand the importance of clearly conveying this information to ensure transparency and ethical integrity in our research.

Ethical Committee Name: The Research Ethics Committee (REC) of the Faculty of Medicine and Pharmacy at Cadi Ayyad University, Marrakech, Morocco. Approval Code: 34/2022 Approval date:27/02/2023

Round 2

Reviewer 1 Report

Comments and Suggestions for Authors

I have no further questions.

Comments on the Quality of English Language

Minor editing of English language required.

Reviewer 2 Report

Comments and Suggestions for Authors

At present form the article can be accepted for publication. Moust of my questions (critical remarks) have been answered.